# Efficacy of Two Entomopathogenic Fungi, *Metarhizium brunneum*, Strain F52 Alone and Combined with *Paranosema locustae* against the Migratory Grasshopper, *Melanoplus sanguinipes*, under Laboratory and Greenhouse Conditions

**DOI:** 10.3390/insects10040094

**Published:** 2019-03-30

**Authors:** Wahid H. Dakhel, Alexandre V. Latchininsky, Stefan T. Jaronski

**Affiliations:** 1Department of Veterinary Pathobiology, College of Veterinary Medicine, University of Missouri, Columbia, MO 65211, USA; 2Food and Agriculture Organization of the United Nations (FAO UN), 00153 Rome, Italy; alexandre.latchininsky@fao.org; 3Department of Agriculture, USDA, Agriculture Research Service (ARS), Sidney, MT 59270, USA; stefan.jaronski@ars.usda.gov

**Keywords:** *Metarhizium brunneum* F52, *Paranosema locustae*, Melanoplus sanguinipes, combination, fungal entomopathogens, biological control, synergistic, laboratory, greenhouse

## Abstract

Grasshopper outbreaks cause significant damage to crops and grasslands in US. Chemical control is widely used to suppress these pests but it reduces environmental quality. Biological control of insect pests is an alternative way to reduce the use of chemical insecticides. In this context, two entomopathogenic fungi, *Metarhizium brunneum* strain F52 and *Paranosema locustae* were evaluated as control agents for the pest migratory grasshopper *Melanoplus sanguinipes* under laboratory and greenhouse conditions. Third-instar grasshoppers, reared in the laboratory, were exposed up to fourteen days to wheat bran treated with different concentrations of each of the fungi alone or the two pathogens combined. In the greenhouse, nymphs were placed individually in cages where they were able to increase their body temperatures by basking in the sun in an attempt to inhibit the fungal infection, and treated with each pathogen alone or in combination. Mortality was recorded daily and presence of fungal outgrowth in cadavers was confirmed by recording fungal mycosis for two weeks’ post-treatment (PT). For combination treatment, the nature of the pathogen interaction (synergistic, additive, or antagonistic effects) was also determined. In laboratory conditions, all treatments except *P. locustae* alone resulted in grasshopper mortality. The application of the pathogen combinations caused 75% and 77%, mortality for lower and higher concentrations, respectively than each of the pathogens alone. We infer a synergistic effect occurred between the two agents. In greenhouse conditions, the highest mortalities were recorded in combination fungal treatments with a *M. brunneum* dose (60% mortality) and with a combination of the two pathogens in which *M. brunneum* was applied at high rate (50%) two weeks after application. This latter combination also exhibited a synergistic effect. Exposure to the *P. locustae* treatment did not lead to mortality until day 14 PT. We infer that these pathogens are promising for developing a biopesticide formulation for rangeland pest grasshopper management.

## 1. Introduction

Grasshoppers (Orthoptera: Acrididae) are important rangeland herbivores that compete with other wildlife and livestock for forage [1,2]. Each year in western United States (US) grasshoppers consume more than 20% of rangeland forages, causing damage estimated at $400 million annually in 1983, close to $1 billion in today’s dollars [3]. There are nearly 500 species of grasshoppers in the 17 US western states and over 100 grasshopper species in Wyoming [4,5]. Of those, about a dozen species are recurrent economic pests including five species in the genus *Melanoplus,* particularly, the migratory grasshopper *M. sanguinipes* (Fabricius) [4,6].

During grasshopper outbreaks, a significant effort is needed to prevent damage to rangelands and crops, which is done by large-scale insecticide applications. For instance, more than 8 million hectares of rangelands and crops were blanketed by five million liters of broad-spectrum chemical insecticides at a cost of $75 million in the mid-1980s [7]. Insecticides are still broadcast in rangelands, with almost 2.5 million hectares of Wyoming rangelands treated with insecticides to manage a severe grasshopper infestation [8].

Currently, chemical pesticides, particularly carbaryl, diflubenzuron and malathion, are used to suppress grasshopper outbreaks in the US [9]. The risks of using chemical pesticides to control grasshopper outbreaks include potential deleterious effects on non-target organisms [10] and the recognized environmental side effects. These risks have encouraged interest in developing fungal pathogens because of their environmentally compatible qualities and reduced impacts on non-target beneficial arthropods including pollinators [11]. Possible uses of biological control agents to reduce use of chemical insecticides are under study in federal grasshopper control programs [12].

In 2005, the insect pathogenic fungus *Metarhizium anisopliae* (Metchnikoff) Sorokin, strain F52, more recently classified as *M. brunneum* F52 (Hypocreales: Clavicipitaceae), was commercialized as a microbial pesticide in the US to control Coleoptera in horticulture and turf management and soft-bodied ticks [12]. *M. brunneum* F52 is now recognized as a pathogen of more than 100 insect species from the Hemiptera, Lepidoptera, Thysanoptera, and Diptera orders [13]. It has led to high mortality against the Mormon cricket, *Anabrus simplex* (Orthoptera: Tettigoniidae), especially under laboratory conditions [12]. The mode of infection can be summarized in the following steps: adhesion, germination, differentiation and penetration. Adhesion involves spore attaching to the cuticle wall, where it germinates to produce an initial hyphal tube (appressorium). This hyphal tube penetrates directly through the exoskeleton and epidermis to reach the hemocoel, in which it develops as yeast-like blastospores [14]. The infection process occurs with the aid of both mechanical and enzymatic degradation of the cuticle. The host will be killed as a result of starvation, nutrient depletion or body obstruction by proliferation of the hyphal bodies [15].

*Paranosema (Nosema) locustae* (Canning), Sokolova (Microsporidia: Nosematidae) was isolated from the African migratory locust, *Locusta migratoria migratorioides* in 1953. Presently, this pathogen is the only microsporidium commercially produced and registered in the US for control of rangeland insects [16,17]. *P. locustae* has a host range of more than 100 orthopteran species and is found across North and South America, Africa, Australia, China, and India [16,18]. The infection of *P. locustae* spores begins in the grasshopper midgut, and then spreads to the fat body. Once these spores are ingested, the infection occurs through the sporoplasm, which is inoculated by the spores through a polar filament into the midgut epithelial cell. Spores will then germinate in the body to affect tissues and hemocytes [19].Compared to chemical insecticides, the *Paranosema* and *Metarhizium* fungal pathogens are slower in killing infected insects [20,21]. To address this shortcoming, the efficacy and/or speed of acridid control may be improved by combining two pathogens [22,23].

Many insect species are able to thermoregulate across changing ambient temperatures [24]. Aside from maintaining optimal thoracic temperatures for locomotion, grasshoppers infected with fungal conidia in the field develop “behavioral fevers”. Many insect orders can generate behavioral fevers, including house flies and stable flies. They assume basking postures or locations that elevate their body temperature above optimal temperatures for microbial growth [25,26]. The mortality rates in field experiments are often lower compared to controlled conditions in a laboratory [11,12]. Performing the experiment in a greenhouse, where the grasshoppers are exposed to sunlight and could thermoregulate, is an intermediate step between laboratory and the field settings. Typically, formulations of the *Metarhizium*-based biopesticides used in acridid control consist of suspensions of fungal spores in oil [20]. *P. locustae* is formulated as solid bait on wheat bran carrier [27]. In the present study, the wheat bran based solid bait formulation was used for both fungal pathogens, *M. brunneum* F52 and *P. locustae*. We tested the hypothesis that despite the potential thermoregulation, the pathogen treatments will yield high mortality levels to consider them as effective grasshopper biocontrol agents. In this paper we report on the outcomes of experiments designed to test our hypothesis.

## 2. Materials and Methods

### 2.1. Experimental Setting

The study was conducted in the Department of Ecosystem Science and Management at the University of Wyoming’s College of Agriculture and Natural Resources insect rearing laboratory and in the greenhouse located at the University of Wyoming Laramie Research and Extension Center, Laramie (41°14′ N, 105°5′ W; elevation 2184 m), Wyoming, USA. In the laboratory, *M. sanguinipes* were reared using a modified protocol of Hinks and Erlandson [28] in which grasshoppers were maintained in cages (Figure 1a) under laboratory conditions at 27 ± 1 °C, 16D:8L photoperiod, and 31 ± 1% relative humidity (RH). *M. sanguinipes* nymphs of a non-diapausing strain from the USDA-ARS-NPARL colony were reared in insect cages (BugDorm-1^®^, MegaView Science Co., Ltd., Taiwan) and provided daily with organic lettuce (Pure Pacific^®^; Romaine) plus wheat bran (ConAgra Mills) as a dietary supplement until they reached the third instar.

For the greenhouse experiment, the nymphs were transferred to a greenhouse where the temperature and relative humidity were kept at 27 ± 1 °C and 33 ± 1% RH, with a natural photoperiod of approximately 16D:8L. One hundred and twenty cylinder-shaped cages (30 cm H × 8 cm D) were constructed of metal cloth (Figure 1b). This size was chosen to fit over sprouted spring wheat (*Triticum aestivum*, Prairie Gold^®^ brand from Wheat Montana Farms) grown in round plastic plant pots (12.5 cm H × 8 cm D). All pots with cages were kept on trays placed on a 1m high growing bench.

### 2.2. Fungal and Baits Preparations

Prior to bait formulation, viability of *M. brunneum* and *P. locustae* was determined. Dry spores were suspended in sterile water and diluted in 0.1% surfactant (Silwet^®^ L-77, Helena Agri Enterprises LLC, Collierville, TN, USA). Conidial viability was determined by placing a dilute suspension of conidia onto potato dextrose agar, incubating at 27 °C for 18 h and then examining the conidia for germination using 650× phase contrast magnification. A minimum of 400 conidia were examined for germination and a conidium was considered viable (germinated) if it had formed a visible germination peg throughout the specified incubation time.

*M. brunneum* F52 conidia from the USDA-ARS-NPARL (Sidney, MT) were applied to wheat bran carrier (ConAgra Mills) or to commercial *P. locustae* bait (NoLo^®^, M&R Durango, Ignacio, CO, USA). The baits were prepared with 100 g of wheat bran spread on a large baking sheet and sprayed using a fine spray (DeVilbiss^®^, Carlisle Fluid Technologies, Scottsdale, AZ, USA) hand atomizer connected to a pressurized air supply, with culinary canola oil at the rate of 3% volume/weight with constant mixing; the canola oil acted as a binder for fungal conidia. The *P. locustae* commercial formulation consisted of wheat bran carrier at 1 × 10^9^ spores per 454 g (EPA registration# 46149-2). The baits were stored for five days at 4 °C prior to use in the assay.

In the laboratory experiment, treatments of *M. brunneum* alone came in two concentrations, low and high, with six replicates each. Therefore, *Metarhizium* was added in quantities sufficient to achieve 0.15 × 10^9^ viable conidia g^−1^ wheat bran (low dose) and 0.49 × 10^9^ viable conidia g^−1^ (high dose). These rates are equivalent to 3.3 × 10^11^ conidia/hectare and 10.78 × 10^11^ conidia/hectare when the baits are delivered at 2.2 kg ha^−1^, a typical dose rate used in grasshopper control [9]. Treated bran was then transferred to a 1L container and the requisite amount of conidial powder added. The container was then sealed and agitated for 10 min to thoroughly coat the bran. Treatment with *P. locustae* alone was another treatment (2.2 × 10^6^
*P. locustae* spores/g) with six replicates.

### 2.3. Treatments Application to the Hosts

In laboratory, ten third instar unsexed *M. sanguinipes* were randomly placed into a 22.86 × 12.7 × 13.97 cm plastic cage^®^ covered with window screen mesh (Figure 1a). The eight experimental treatments were applied to a total of 48 cages (480 grasshoppers). Each treatment was replicated 6 times with biologically independent replicates. The caged nymphs were exposed to continuous, long-term exposure (LTE) with treated bait for 14 day (d), or short-term exposure (STE) for 3 d.

Treatments were applied as two formulations: single pathogen (individual treatment) or mixed pathogens (combination). Applications of *M. brunneum* alone were delivered to caged grasshoppers for 3 d at low concentration (0.15 × 10^9^ conidia/g; 3DM1) and high concentration (0.49 × 10^9^ conidia/g; 3DM2), and then substituted with untreated wheat bran for remaining 11 d of the experiment (STE). Besides STE, the LTE (14 d) was applied as low 0.15 × 10^9^ conidia/g (EDM1) and high 0.49 × 10^9^ conidia/g (EDM2) concentrations. Application of *P. locustae* bait was fed to grasshoppers for 3 d a single dose rate of 2.2 × 10^6^ spores/g (N), and then substituted with untreated wheat bran for 11 d.

Combined *M. brunneum* and *P. locustae* treatments were set in in two concentrations, low 0.15 × 10^9^ conidia/g (CMN1) and high 0.49 × 10^9^ conidia/g (CMN2), with six replicates each. In these combinations, the dose rate of *P. locustae* remained the same as in the single pathogen treatment, equaling 2.2 × 10^6^ spores/g. The combination treatments were provided to the grasshoppers for STE (3 d), and then substituted with untreated wheat bran for the remaining 11 d. Untreated wheat bran was provided continuously as the control (C) to groups of nymphs. The details of all eight treatments and dose rates are presented in Table 1. All 48 cages were placed randomly in environmental chambers. All treatments, including untreated wheat bran, were delivered to grasshopper populations in plastic square weighing dishes (89 mm × 89 mm × 25 mm) as 31 flakes of formulation bait per cage (Figure 1a). This amount was based on a standard bait application rate of 2.2 kg ha^−1^ extrapolated to the area under each cage.

In the greenhouse, one unsexed grasshopper was caged and received six flakes of each of the six different formulations of pathogen-treated wheat bran bait, and those flakes were applied directly on the bottom of each pot once at the start of experiment. Each treatment was replicated 20 times. Over a two-week period, nymphs had continuous access to the bran bait and wheat tillers. The rate of bait (0.0011 g, which is equivalent to six flakes/cage), was calculated taking into account the area of the cage (50.3 cm^2^) and the standard rate for *P. locustae* grasshopper bait recommended by the Nolo^®^ bait company, equivalent to 2.2 kg of bait ha^−1^. In each of the 120 cages, a single unsexed third-instar *M. sanguinipes* nymph was placed for continuous exposure to different mixture rates of *M. brunneum* F52 and *P. locustae* similar to the laboratory experiment. The treatments included EDM1, EDM2, CMN1, CMN2, N, and Control (C).

### 2.4. Data Collection

Grasshopper mortality was registered daily for 16 d. Dead grasshoppers as well as grasshopper frass, were removed from the cages daily to avoid scavenging by grasshoppers that were still alive. Collected cadavers were washed in 0.5% NaOCl for 1 min and rinsed in distilled water and placed in Petri dishes containing a cotton ball moistened with distilled water to provide high humidity and promote fungal outgrowth in the cadaver. The Petri plates were then incubated at 27 °C in order to observe fungal mycosis on dead grasshoppers [29] (Figure 2).

### 2.5. Statistical Analysis

Both experiments were laid out as a randomized complete block design with six replicates under laboratory and twenty replicates under greenhouse conditions. Data of dead grasshoppers were collected at 7, 10 and 14 d after treatment using a one factor factorial analysis of variance (ANOVA) [30]. Post-hoc tests were performed using Fisher’s protected LSD (α = 0.05) procedure, and calculated following Milliken and Johnson method [31]. Homoscedasticity of variances was tested using Hartley’s ($s_{max}^{2}/s_{min}^{2}$) statistic and, if the variances were found to be heteroscedastic, the analysis was performed using ($1/\sqrt{s_{i}^{2}}$) as weights, where $s_{i}^{2}$ was the variances of the $i^{th}$ treatment. Normality of residuals was assessed by the Shapiro-Wilk test and normal probability plots. Statistical computations for normality were facilitated using the UNIVARIATE procedure of SAS [32]. Virulence of the pathogen treatments was evaluated using the median survival times (MST) calculated from Kaplan and Meier survivorship analysis 16 d post-treatment (PT) [33]. Combination treatments (*M. brunneum* + *P. locustae*) were analyzed at 14 d PT for additive, antagonistic or synergistic interactions using bioassay method one of Nishimatsu and Jackson [34]. First, the expected percentage mortality was calculated using the formula *P_e_* = *P*_0_ + (1 − *P*_0_) (*P*_1_) + (1 − *P*_0_) (1 − *P*_1_) (*P*_2_) where, *P_e_* = expected percentage mortality from combined treatments divided by 100, *P*_0_ = percentage control mortality divided by 100, *P*_1_ = percentage mortality from agent 1 divided by 100; and *P*_2_ = percentage mortality from agent 2 divided by 100. Second, a Chi-square statistic was calculated using the formula χ^2^ = (*L*_0_ − *L_e_*)^2^/*L_e_* + (*D*_0_ − *D_e_*)^2^/*D_e_* where, *L*_0_ = Observed # living larvae, *L_e_* = Expected # living larvae, *D*_0_ = Expected # dead larvae, *D_e_* = Observed # dead larvae. Finally, the determination of the effect (antagonism, additivity or synergism) was done based on the hypothesis with 1 df and *α* = 0.05. If χ^2^ < 3.84 additivity is indicated. If χ^2^ > 3.84, and *P_c_* < *P_e_* (*P_c_* = observed mortality of combination, and *P_e_* = expected mortality from combined treatments) antagonism is indicated. If χ^2^ > 3.84 and *P_c_* > *P_e_*, synergism is indicated. If *p* ≤ 0.05 the observed mortality differs from the expected in at least two treatments. However, if *p* ≥ 0.05 the observed mortality is the same as the expected over all treatments.

Further analysis was performed on greenhouse experiment. Data were subjected to a two dimensional χ^2^ test. As the number of grasshoppers per treatment was fixed, the appropriate sampling model for these analyses was that of a product-multinomial [35], with the null hypotheses being the ratio of dead: alive grasshoppers is the same among the treatments versus the alternate that at least two were different (α = 0.05). If the null hypothesis was rejected and the alternate was accepted, then all possible pair-wise 2 × 2 χ^2^ tests were conducted to determine which treatments were similar and which were different. Finally, as the number of grasshoppers that were dead represented a response variable (and not an explanatory variable) [35], results of the pair-wise tests were summarized as % mortality, along with their observed dead: alive ratios, for each of the three times. Statistical computations were facilitated using the FREQ procedure of SAS [32]. Over a period of 16 days, analysis of speed of lethal action was conducted to compare the virulence between the combination treatments and the single fungus treatments. Survival analysis was conducted using MST, and all treatments that caused more than 50% of grasshopper mortality were analyzed using Kaplan–Meier method and log-rank tests [33].

## 3. Results

Cumulative mortality of *M. sanguinipes* nymphs subject to various treatments at the laboratory and greenhouse is shown in Figure 3.

### 3.1. Treatments Exposure in the Laboratory

Analysis of variance indicated a significant differences occurred between at least two treatments at d 14 (df = (7, 40), *F* = 120.99, *p* < 0.0001; Table 2). All experimental treatments except 3DM1 were significantly different from the control. Additionally, a significant effect of grasshopper mortality was observed in treatment of EDM2, which differed significantly when compared to 3DM1, EDM1 and 3DM2.

#### 3.1.1. Single Pathogen Treatments

In the laboratory, at short-term (3 d) exposure to *M. brunneum* (STE), mortality from the low-dose (low conidial concentration) treatment (3DM1) was consistently lower than mortality from the high dose (high conidial concentration, 3DM2) at 7, 10 and 14 d PT (Table 2). Similarly, high-dose of *M. brunneum* treatments at continuous exposure (LTE) showed high mortality, with significant difference between high-dose (EDM2) and low-doses (EDM1) of *M. brunneum* at 10 and 14 d PT (Table 2). The treatment with a high-dose, continuous exposure to *M. brunneum* (EDM2) resulted in the highest numerical grasshopper mortality (87.3%) in the entire experiment 14 d after treatment.

Mortality of *M. sanguinipes* nymphs exposed to *M. brunneum* treatments for only 3 d (STE; 3DM1) was significantly lower than the mortality from LTE-low *M. brunneum* concentration (EDM1) at 7, 10 and 14 d PT. Similarly, mortality after STE-high *M. brunneum* concentration (3DM2) was significantly lower than the mortality after LTE-high *M. brunneum* concentration (EDM2) at 7, 10 and 14 d PT. The other *M. brunneum* treatments (3DM2, EDM1 and EDM2) produced mortalities statistically different compared to that observed in the control (C) (Table 2).

During most of the observation period, *P. locustae* did not cause mortality of *M. sanguinipes* nymphs and only on the 16th d death was first observed on grasshoppers treated with this pathogen (Table 2).

#### 3.1.2. Combination of Treatments

At 10 and 14 d PT, but not 7 d, the grasshopper mortality in the combination treatment with low dose of *M. brunneum* CMN1 resulted in mortalities statistically different from untreated control (Table 2). At two earlier dates (7 and 10 d after application in the laboratory) the combination treatment with high dose of *M. brunneum* (CMN2) showed high significant mortality compared to the combination treatment with low (CMN1) dose of *M. brunneum* (Table 2). Overall, the combination treatments resulted in cumulative mortalities over 70% two weeks after application in the laboratory (Table 2).

### 3.2. Treatments Exposure in the Greenhouse

A χ^2^ analyses at 7, 10, and 14 days’ post initial exposure indicated there were differences in dead: alive ratios among the six treatments at each assessment date (7 days: $\chi_{5}^{2}$ = 15.968, *p* = 0.0069; 10 days: $\chi_{5}^{2}$ = 23.2926, *p* = 0.0003; 14 days: $\chi_{5}^{2}$ = 24.3389, *p* = 0.0002). Subsequent comparisons among treatments indicated that for 7-day continuous exposure time, mortality from the *M. brunneum* at high dosage + *P. locustae* treatment was significantly greater than mortality from the *M. brunneum* alone treatment at low dosage, as well as the mortality that occurred from both the untreated control and the *P. locustae* alone treatment (Table 2). In addition, the mortality that occurred from the *M. brunneum* + *P. locustae* treatment at low dosage and the *M. brunneum* alone treatment at high dosage was significantly greater than the mortality that occurred from both the untreated control and the *P. locustae* alone treatments.

Results from comparisons at 10-day continuous exposure were identical to those observed at the 7-day continuous exposure time with the exception that the mortality occurring from the *M. brunneum* at low dosage + *P. locustae* treatment was significantly greater than the mortality that occurred from the *M. brunneum* alone treatment at low dosage, the untreated control treatment, and the *P. locustae* alone treatment (Table 2).

Results from comparisons made at the 14-day continuous exposure had mortality from the *M. brunneum* + *P. locustae* treatment at both dosages being significantly greater than the mortalities that occurred from the *M. brunneum* treatment alone at low dosage, the untreated control, and the *P. locustae* alone treatment (Table 2). In addition, mortality occurring from the *M. brunneum* alone treatment at high dosage was significantly greater than the mortality that occurred at both the untreated control and the *P. locustae* alone treatment (Table 2).

Finally, it bears mentioning that mortalities that occurred from the *M. brunneum + P. locustae* treatments at both dosages plus the mortalities that occurred from the *M. brunneum* alone treatment at high dosage were not significantly different, regardless of the duration of exposure time (up to and including 14 days). Likewise, mortalities that occurred from the *M. brunneum* alone treatment at low dosage, the untreated control, and the *P. locustae* alone treatments, were not significantly different. Lastly, mortalities from the *M. brunneum* alone treatment at high dosage were not significantly different from the *M. brunneum* alone treatment at low dosage (Table 2).

### 3.3. Virulence

The laboratory results of the virulence bioassay, in which the median survival time (MST) for third-instar nymphs of *M. sanguinipes* was assessed at 16 d are presented in Table 3. Numerically, median survival time of grasshoppers treated with LTE-high concentration of fungus EDM2 was 8.2 d. Similar to EDM2 treatment, the MST for the combination treatment at high concentration CMN2 had the same length of 8.2 d. Because mortalities of grasshoppers over 16 d were less than 50% (Table 2), the MST for *P. locustae* alone, untreated control, and STE-low *M. brunneum* concentration (3DM1) treatments could not be determined. Cox Proportional Hazards analysis [36] with daily observation data showed that fungal treatments with a high concentration of EDM2 and CMN2 were statistically similar in their virulence (Z = −0.84, *p* = 0.4013, Relative Risk of death (Rel. Risk) = 0.92 for α = 0.05). At the same time, the two treatments were significantly more virulent than control (Table 4).

In the greenhouse, survival analysis showed that the effect of combined treatments with a mixture of two fungi resulted in the highest mortality, thus their median survival time (MST) was determined at 16 days PT in Table 3. Numerically, killing speed of grasshopper nymphs with the combination at low concentration (CMN1) was 10.1 days. In addition, fungus combinations at high concentration (CMN2) had MST of 13.5 days. Thus, based on log rank test (α = 0.05), the virulence of CMN1 and CMN2 were not significantly different (Z = −0.30, *p* = 0.7654, Relative Risk of death (Rel. Risk) = 0.88) (Table 4). However, virulence in both of combination fungal treatments were statistically different when compared to untreated control (Table 4). MST for other fungal treatments was not calculated for survival analysis because nymphal mortality was below 50% (Table 4).

### 3.4. Interactions between the Pathogens

Results of the analysis of the interactions between the pathogens at the laboratory and greenhouse are presented in Table 5. In the laboratory, combination treatment of *M. brunneum* and *P. locustae* together at high concentration versus LTE-high concentration of *M. brunneum* and *P. locustae* alone (CMN2 vs. EDM2, N), the observed mortality in the pathogen combination treatment was lower than the expected based on the separate performance of each pathogen. This means that in this combination, the pathogens exhibited antagonism. In three other cases, the observed mortality from the combination treatments exceeded the expected mortality. Out of those, in one case, comparing combination treatment of *M. brunneum* and *P. locustae* together at low concentration versus STE-low of *M. brunneum* and *P. locustae* alone (CMN1 vs. 3DM1, N), the pathogen combination produced a synergistic effect while in two others (CMN1 vs. EDM1, N and CMN2 vs. 3DM2, N), the combination resulted in additivity (Table 5).

For the greenhouse, at two weeks post treatment (PT), the combination of the two entomopathogenic fungi *M. brunneum* F52 and *P. locustae* at low rate (CMN1) resulted in a synergistic effect (i.e., Supplemental effect that is greater than the sum of each effects alone) when compared to *M. brunneum* F52 at low rate (EDM1) and *P. locustae* alone (N) treatments (*P_c_* = 60%, *Pe* = 31.6%, χ^2^ = 7.463, df = 1; Table 5). Additive effect (an effect that occurs when the two agents of insect control act independently of each other) was observed with CMN2 treatment when compared to EDM2 and N alone (*P_c_* = 50%, *Pe* = 53.0%, χ^2^ = 7.463, df = 1; Table 5). No antagonistic effects between the two fungal treatments were detected.

## 4. Discussion

In this study, two fungal pathogens, *M. brunneum* F52 and *P. locustae* were tested either singly or together under co-infection, delivered on wheat bran carrier, to control 3rd instar nymphs of the grasshopper *M. sanguinipes*. When applied alone, the *M. brunneum* caused high mortality of tested insects, especially at high dose 3-d exposure and continuous exposure (3DM2; EDM2). Such treatments resulted in 68 to 88% mortality 14 d after application, respectively. This effect was likely due to production of insecticidal toxins (destruxins) that impair host metabolism and the development of the fungal mycelia, which consume the host nutrients in the hemolymph [37]. Our findings are consistent with the results of Foster et al. [12] who also found high effects of *M. brunneum* F52 against Mormon crickets, *A. simplex*, under laboratory conditions.

In contrast, grasshoppers subject to a 3-d exposure to *P. locustae* did not exhibit mortality until the 16th day PT, which is consistent with previous studies. In Africa, *P. locustae* was evaluated in the laboratory against first, second, and third-instar nymphs of Senegalese grasshopper, *Oedaleus senegalensis* [38]. There was higher mortality of early instars that eventually died as fourth-instar nymphs or became moribund as fifth instars at 30 d of observation. There are several potential explanations why the single microsporidian pathogen did not lead to the disease. In terms of pathology, microsporidiosis is categorized as a chronic disease, meaning the disease is slow-acting instead of producing an acute, or fast-acting, infection [39]. Previously, a study concluded that *P. locustae* needs ample time to invade insect tissues and proliferate [40]. This is similar to the findings of the present study, which indicate that mortality of immature grasshoppers caused by feeding on *P. locustae* alone occurred only at d 16. Additionally, low mortality rate of grasshoppers being treated with bait consisting of just *P. locustae* might be explained by grasshoppers either not consuming the microsporidia-treated bait or avoiding it altogether, a behavior which has been sometimes observed in laboratory and field studies [12]. *P. locustae* alone acts very slowly, produces low and inconsistent mortality among host grasshoppers and therefore should not be considered as a viable microbial control agent.

However, when *P. locustae* and *M. brunneum* were applied together, the combination two-pathogen led to additive or even synergistic effects two weeks after treatment. The synergistic reaction occurred, somewhat unexpectedly, when low (and not high) doses of *M. brunneum* were used (CMN1). Such treatment resulted in ten-fold increase in mortality compared to that produced by *M. brunneum* by itself. At high dose rates (CMN2), the combination effect resulted only in one- fold increase in grasshopper mortality, compared to that produced by *M. brunneum* by itself [41]. A plausible explanation is that *P. locustae* infection develops primarily in midgut and then migrates to fat body. This may impair host metabolism and energy storage, resulting in sluggish behavior [39]. This could also cause the infected insects to become stressed, which favors the second infection by *M. brunneum* F52.

A study investigated the effects of simultaneous and individual administration of *P. locustae* and *M. anisopliae var. acridum* (known as *M. acridum*) to fifth-instar nymphs of desert locust, *Schistocerca gregaria*, under laboratory conditions [22]. As a carrier, they used solid wheat bran bait for 21 d. They recorded synergistic effects in treatments of the highest concentration of *M. acridum*, thus confirming our result that combined pathogens killed grasshoppers faster and more efficiently than either pathogen on its own. However, we found a synergistic interaction when the two pathogens were applied at lower rates, which is more advantageous economically and environmentally. Our findings support the results of Ericsson et al. [42] who demonstrated that dual infections consisting of the bioinsecticide spinosad and biological agent *M. anisopliae* caused more in reductions in wireworm populations, *Agriotes* spp. (Coleoptera: Elateridae) than either treatment by itself. Our results indicate that combining even a low dose rate of *M. brunneum* with *P. locustae* on wheat bran bait increases the pathogenicity and rate of mortality in host grasshoppers.

Although the fungal pathogen such as *M. brunneum* does not need to be ingested to cause a mycosis, it was nevertheless quite efficacious against grasshoppers that consumed the wheat bran bait with fungal conidia. We infer the infection occurs via conidia penetration through the insect cuticle on mouthparts as suggested by Hajek and St. Leger [37].

Our study was performed under controlled conditions of a laboratory. Numerous environmental factors may influence the performance of the microbiological entomopathogens under the field conditions [12]. One of the most important is solar UV radiation, which may kill the pathogen conidia directly or provoke a behavioral response in grasshoppers allowing them to increase their body temperature and overcome the infection [43]. Therefore, the results of the greenhouse experiment showed that, despite the assumed host thermoregulation, certain pathogen treatments caused grasshopper mortality under greenhouse conditions. For example, a high rate of *M. brunneum* application resulted in 45% mortality 14-day PT. Conversely, the application of the low *M. brunneum* rate yielded only 20% mortality after the same length of incubation (Table 2). Grasshoppers treated with *P. locustae* did not die until 14-day PT, understanding their mortality was lower than in untreated control. These results are consistent with data obtained in the laboratory experiments as well as in previous studies [44,45].

Combinations of the two pathogens resulted in mortality rates of 50% or higher at 14-day PT. As expected, these mortalities were lower than those obtained under the controlled laboratory conditions (close to 80%, Table 2). Also, the pathogens acted slower: the first signs of mortality appeared 5 and 6-day PT versus 3-day PT in the lab. Yet the fact that at 14-day PT the two combination treatments yielded more than 50% mortality allows consideration of the combination of *M. brunneum* and *P. locustae* as promising for developing an operational biological control formulation against pest grasshoppers, most of whom died of mycosis, despite behavioral fevers.

The simulated field study demonstrated that, applied alone, *M. brunneum* was quite efficacious at a high rate despite the fact that the tested grasshoppers were presumably able to thermoregulate. This suggests that possibly, *M. brunneum* exhibits tolerance to high body temperatures of grasshoppers, is an important trait of a potential microbiological control agent. Compared to another entomopathogenic fungus *Beauveria bassiana*, fungi from the genus *Metarhizium* appear more virulent against grasshoppers *Schistocerca americana* and *Melanoplus sanguinipes* even after 21 d of thermoregulation regime [46]. Previous research findings suggest that Mormon cricket *Anabrus simplex* Haldeman (Orthoptera: Tettigoniidae) also can be affected by *M. brunneum*. Even though Mormon cricket produces body heat of 39–40 °C during thermoregulation, *M. brunneum* achieved consistent and satisfactory efficacy, especially under laboratory conditions [12].

## 5. Conclusions

The application of two fungal pathogens, *M. brunneum* F52 and *P. locustae* formulated on wheat bran bait proved effective against migratory grasshoppers *M. sanguinipes* nymphs under both laboratory and greenhouse conditions. This fungal pathogen combination showed promise in developing a biopesticide against pest grasshoppers. Further research is required to evaluate the efficacy, virulence and persistence of this perspective pathogen combination in the field.

## Figures and Tables

**Figure 1 insects-10-00094-f001:**
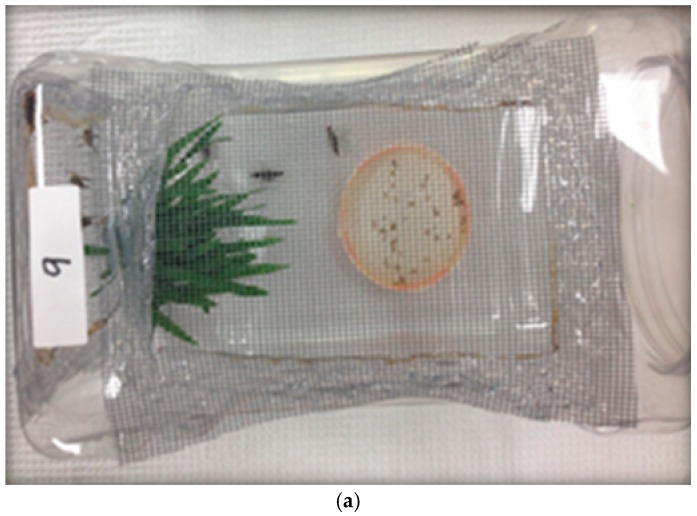
Plastic cage^®^ covered with window screens mesh in the laboratory (**a**); cylindrical metal hardware cloth cages over the sprouted wheat in the greenhouse (**b**).

**Figure 2 insects-10-00094-f002:**
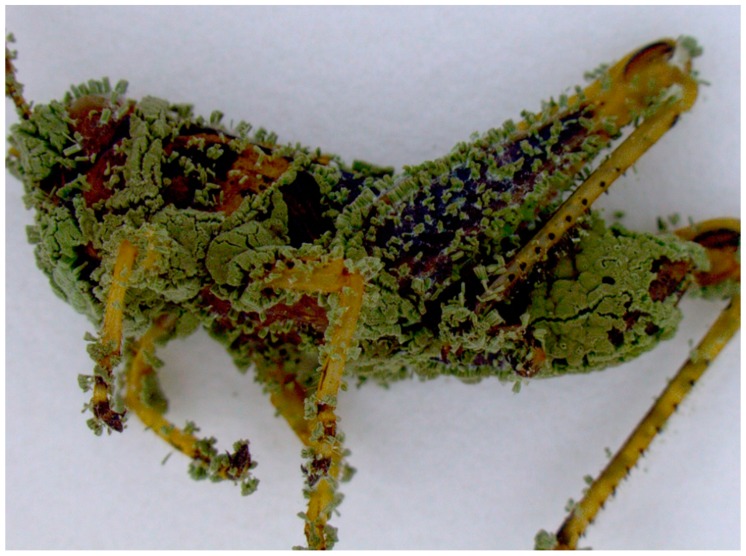
*M. brunneum* F52 sporulation on the body of grasshopper *M. sanguinipes*.

**Figure 3 insects-10-00094-f003:**
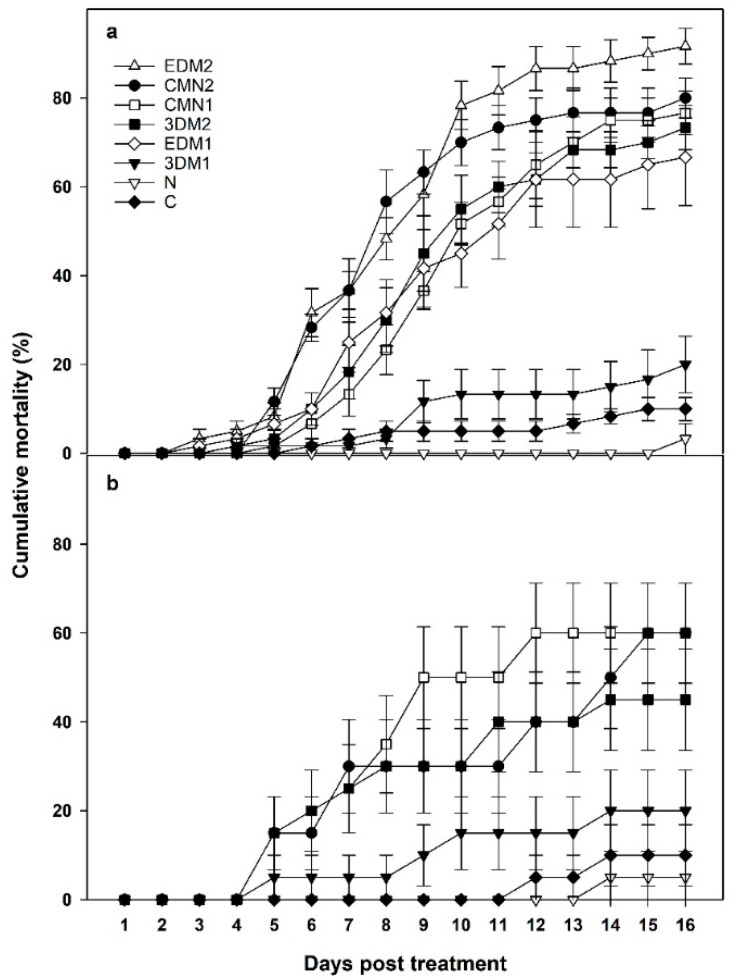
Cumulative mortality (%SE) of laboratory (**a**) and greenhouse (**b**) *Melanoplus sanguinipes* nymphs treated with *Metarhizium brunneum* F52 and *Paranosema locustae*. 3DM1 = 3 days’ exposure (STE) to *M. brunneum* at low concentration. 3DM2 = 3 days’ exposure (STE) to *M. brunneum* at high concentration. EDM1 = Continuous exposure (LTE) to *M. brunneum* at low concentration. EDM2 = Continuous exposure (LTE) to *M. brunneum* at high concentration. CMN1 = 3 days’ exposure (STE) to *M. brunneum* and *P. locustae* at low concentration. CMN2 = 3 days’ exposure (STE) to *M. brunneum* and *P. locustae* at high concentration. N = 3 days’ exposure to *P. locustae*. C = Untreated control.

**Table 1 insects-10-00094-t001:** Materials and rates of baits for treatment applied in study of 3rd instar grasshopper control. These rates were made according to 2.2 kg ha^−1^.

Treatment ^†^	Material	Rate
3DM1	Feeding low concentration of *M. brunneum* F52 for only 3 days	0.15 × 10^9^ conidia g^−1^
3DM2	Feeding high concentration of *M. brunneum* F52 for only 3 days	0.49 × 10^9^ conidia g^−1^
EDM1	Feeding low concentration of *M. brunneum* F52 continuously for 14 days	0.15 × 10^9^ conidia g^−1^
EDM2	Feeding high concentration of *M. brunneum* F52 continuously for 14 days	0.49 × 10^9^ conidia g^−1^
CMN1	Feeding low concentration of both *M. brunneum* F52 and *P. locustae* for only 3 days	0.15 × 10^9^ spores g^−1^
CMN2	Feeding high concentration of both *M. brunneum* F52 and *P. locustae* for only 3 days	0.49 × 10^9^ spores g^−1^
N	Exposure grasshoppers for *P. locustae* by itself	2.2 × 10^6^ spores g^−1^
C	Untreated control	Untreated control

^†^ 3DM1 = 3 days’ exposure (STE) to *M. brunneum* at low concentration. 3DM2 = 3 days’ exposure (STE) to *M. brunneum* at high concentration. EDM1 = Continuous exposure (LTE) to *M. brunneum* at low concentration. EDM2 = Continuous exposure (LTE) to *M. brunneum* at high concentration. CMN1 = 3 days’ exposure (STE) to *M. brunneum* and *P. locustae* at low concentration of *M. brunneum*. CMN2 = 3 days’ exposure (STE) to *M. brunneum* and *P. locustae* at high concentration of *M. brunneum*. N = 3 days’ exposure to *P. locustae*. C = Untreated control.

**Table 2 insects-10-00094-t002:** Mean percentage of observed mortality of third-instar nymphs of *Melanoplus sanguinipes* after exposure to bait treated with wheat bran of *Metarhizium brunneum* F52 and *Paranosema locustae* for 7, 10, 14 days in laboratory and greenhouse conditions.

Treatment ^†^	7 Days		10 Days		14 Days	
	Laboratory	Greenhouse	Laboratory	Greenhouse	Laboratory	Greenhouse
EDM2	37 ^a^	25 ^ab^	78 ^a^	30 ^ab^	88 ^a^	45 ^ab^
CMN2	37 ^a^	30 ^a^	70 ^ab^	30 ^ab^	77 ^ab^	50 ^a^
CMN1	13 ^bc^	25 ^ab^	52 ^c^	50 ^a^	75 ^ab^	60 ^a^
3DM2	18 ^b^	--	55 ^bc^	--	68 ^b^	--
EDM1	25 ^ab^	5 ^b,c^	45 ^c^	15 ^bc^	62 ^b^	20 ^bc^
3DM1	2 ^d^	--	13 ^d^	--	15 ^c^	--
C	3 ^cd^	0 ^c^	5 ^de^	0 ^c^	8 ^c^	10 ^c^
N	0 ^d^	0 ^c^	0 ^e^	0 ^c^	0 ^d^	5 ^c^

^†^ 3DM1 = 3 days’ exposure (STE) to *M. brunneum* at low concentration (0.15 × 10^9^ conidia g^−1^). 3DM2 = 3 days’ exposure (STE) to *M. brunneum* at high concentration (0.49 × 10^9^ conidia g^−1^). EDM1 = Continuous exposure (LTE) to *M. brunneum* at low concentration (0.15 × 10^9^ conidia g^−1^). EDM2 = Continuous exposure (LTE) to *M. brunneum* at high concentration (0.49 × 10^9^ conidia g^−1^). CMN1 = 3 days’ exposure (STE) to *M. brunneum* and *P. locustae* at low concentration of *M. brunneum* (0.15 × 10^9^ spore g^−1^). CMN2 = 3 days’ exposure (STE) to *M. brunneum* and *P. locustae* at high concentration of *M. brunneum* (0.49 × 10^9^ spore g^−1^). N = 3 days’ exposure to *P. locustae* (2.2 × 10^6^ spores g^−1^). C = Untreated control. Within column, means followed by the same letter are not statistically different at *p* ≤ 0.05.

**Table 3 insects-10-00094-t003:** Median survival time (MST) in days of third-instar nymphs of the migratory grasshoppers, *Melanoplus sanguinipes*, treated with different concentrations of *Paranosema locustae* and *Metarhizium brunneum* F52 for 16 days in laboratory and greenhouse conditions.

Treatment ^†^	MST(days) ^§^		Lower 95%		Upper 95%	
	Laboratory	Greenhouse	Laboratory	Greenhouse	Laboratory	Greenhouse
EDM2	8.2	--	7.2	--	10.3	--
CMN2	8.2	13.5	7.1	8.4	9.2	18.4
CMN1	10.5	10.1	9.1	7.3	11.9	12.8
3DM2	10.3	--	9.1	--	11.5	--
EDM1	10.7	--	9.6	--	11.9	--

^§^ MST = Median Survival Time. ^†^ 3DM1 = 3 days’ exposure (STE) to *M. brunneum* at low concentration. EDM2 = Continuous exposure (LTE) to *M. brunneum* at high concentration. CMN2 = 3 days’ exposure (STE) to *M. brunneum* and *P. locustae* at high concentration of *M. brunneum*. CMN1 = 3 days’ exposure (STE) to *M. brunneum* and *P. locustae* at low concentration of *M. brunneum*. 3DM2 = 3 days’ exposure (STE) to *M. brunneum* at high concentration. EDM1 = Continuous exposure (LTE) to *M. brunneum* at low concentration.

**Table 4 insects-10-00094-t004:** Comparing different treatments in laboratory and greenhouse using Cox’s Proportional Hazards model based on the survivorship of third-instar grasshopper nymphs (α = 0.05).

Variables ^†^	Coefficient	Standard Error	Z ^£^	*p*	R.R ^¥^
Laboratory					
EDM2 vs. CMN2	−0.08449	0.10067	−0.84	0.4013	0.92
EDM2 vs. untreated (C)	1.04220	0.15864	6.57	0.0000	2.84
CMN2 vs. untreated (C)	0.54848	0.09494	5.78	0.0000	1.73
Greenhouse					
CMN1 vs. CMN2	−0.12217	0.40944	−0.30	0.7654	0.88
CMN1 vs. untreated (C)	2.18975	0.76731	2.85	0.0043	8.93
CMN2 vs. untreated (C)	1.04986	0.38277	2.74	0.0061	2.86

^†^ EDM2 = Continuous exposure (LTE) to *M. brunneum* at high concentration. CMN2 = 3 days’ exposure (STE) to *M. brunneum* and *P. locustae* at high concentration. C = Untreated control. CMN1 = exposure to *M. brunneum* and *P. locustae* at low concentration. Z^£^ = Covariate (Coefficient/Standard error). ^¥^ R.R = Relative Risk of death.

**Table 5 insects-10-00094-t005:** Interactions between treatments applied to third-instar nymphs of *Melanoplus sanguinipes* for 14 days in laboratory and greenhouse conditions.

Treatment ^†^	*P_c_*% *^ζ^*	*P_e_*% ^§^	χ^2^	Interaction
Laboratory				
CMN1 vs. 3DM1, and N	75	22	97.838	Synergism
CMN2 vs. 3DM2, and N	77	71	0.965	Additive
CMN1 vs. EDM1, and N	75	65	2.691	Additive
CMN2 vs. EDM2, and N	77	89	9.909	Antagonism
Greenhouse				
CMN1 vs. EDM1, and N	60	32	7.463	Synergism
CMN2 vs. EDM2, and N	50	53	0.071	Additive

^†^ 3DM1 = 3 days’ exposure (STE) to *M. brunneum* at low concentration. 3DM2 = 3 days’ exposure (STE) to *M. brunneum* at high concentration. EDM1 = Continuous exposure (LTE) to *M. brunneum* at low concentration. EDM2 = Continuous exposure (LTE) to *M. brunneum* at high concentration. CMN1 = 3 days’ exposure (STE) to *M. brunneum* and *P. locustae* at low concentration of *M. brunneum*. CMN2 = 3 days’ exposure (STE) to *M. brunneum* and *P. locustae* at high concentration of *M. brunneum*. N = 3 days’ exposure to *P. locustae*. C = Untreated control. χ^2^ = Chi square. ***^ζ^***
*Pc* = observed percentage mortality from CMN1/CMN2 vs. ^§^
*Pe* = expected percentage mortality from 3DM1/3DM2/EDM1/EDM2, N, and Control treatments.

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
