# Peer review of "Efficacy of Two Entomopathogenic Fungi, Metarhizium brunneum, Strain F52 Alone and Combined with Paranosema locustae against the Migratory Grasshopper, Melanoplus sanguinipes, under Laboratory and Greenhouse Conditions"

_insects, 2019, doi:10.3390/insects10040094_

Round 1
Reviewer 1 Report
Line 21- third instar indicates that the insect is a nymp; nymph is redundant.
Line 24- “generate fever” this needs a short description, general readers will not be familiar with this behavior.
Line- suggest using management instead of control.
Lines 69-76. Suggest a explanation of the difference in mode of action between the two pathogens even though they both are taxinomically fungi.
Line 129- explain why no viability tests were conducted for P. locustae, these formulations have tendencies to lose viability during storage.
Line 139- see line 21 comment.
Lines 145-151- were the nymphs exposed once and then observed for x days or exposed to fresh pathogens daily?
Line 152- delete “in”.
line 193- data were collected at 7..... not analyzed.
Line 232- place a) and b) the left of respective graphs.
Line 273- not when they started to die, when death was first observed.
Line 280- add in the laboratory after application.
Line 305- “were always the same”, replace with “not significantly different”.
Line 309- did you record instar stage when taking mortality data? Reduced growth would be attributed to either pathogens. Also, did you do a postmortem to determine if nymphs had an infection of P. locustae?
Line 410- any reference for this?
In the discussion you mention possibility of thermoregulation, did you ever observe such?
Did you have any indication that nymphs contaminated themselves in addition to consumption of conditions?
Author Response
Responses to Reviewer 1
Line 21- third instar indicates that the insect is a nymp; nymph is redundant.
Authors: Changed were incorporated in the manuscript as suggested.
___________________________________________________________________________
Line 24- “generate fever” this needs a short description, general readers will not be familiar with this behavior.
Authors: Changed were incorporated in the manuscript as suggested.
_______________________________________________________________________________
Line- suggest using management instead of control.
Authors: Changed were incorporated in the manuscript as suggested.
_______________________________________________________________________________
Lines 69-76. Suggest a explanation of the difference in mode of action between the two pathogens even though they both are taxinomically fungi.
Authors: Changed were incorporated in the manuscript as suggested.
______________________________________________________________________________
Line 129- explain why no viability tests were conducted for P. locustae, these formulations have tendencies to lose viability during storage
Authors: The viability test was done for both pathogens. Changed were incorporated in the manuscript as suggested.
Line 139- see line 21 comment.
Authors: Changed were incorporated in the manuscript as suggested.
______________________________________________________________________________
Lines 145-151- were the nymphs exposed once and then observed for x days or exposed to fresh pathogens daily?
Authors: Short treatment exposure (STE) means that the nymphs were exposed to the treatment for 3 days and then mortality observed till day 14 of the experiment. We put the bait in the cages in the first day and take it out of the cages after 3 days then observed mortality. Changed were incorporated in the manuscript as suggested. The long treatment exposure (LTE) means that the grasshoppers were exposed to the bait every day since the first day and left it in the cages for 14 days.
_______________________________________________________________________________
Line 152- delete “in”.
Authors: Changed were incorporated in the manuscript as suggested.
_______________________________________________________________________________
line 193- data were collected at 7..... not analyzed.
Authors: Changed were incorporated in the manuscript as suggested.
________________________________________________________________________________
Line 232- place a) and b) the left of respective graphs.
Authors: Changed were made in figure 3 as suggested.
_________________________________________________________________________________
Line 273- not when they started to die, when death was first observed.
Authors: Changed were incorporated in the manuscript as suggested.
_________________________________________________________________________________
Line 280- add in the laboratory after application.
Authors: Changed were incorporated in the manuscript as suggested.
__________________________________________________________________________________
Line 305- “were always the same”, replace with “not significantly different”.
Authors: Changed were incorporated in the manuscript as suggested.
_________________________________________________________________________________
Line 309- did you record instar stage when taking mortality data? Reduced growth would be attributed to either pathogens. Also, did you do a postmortem to determine if nymphs had an infection of P. locustae?
Authors: We recorded the instar stage when we started the experiment which is third instar. We did not record the reduced growth, we just confirmed the infection. P.locuse did not cause any mortalities (lab) even after 14 days so I did not do a postmortem.
__________________________________________________________________________________
Line 410- any reference for this?
Authors: Changed were incorporated in the manuscript and reference added as suggested.
_________________________________________________________________________________
In the discussion you mention possibility of thermoregulation, did you ever observe such?
Authors: Yes we observed that during the experiment.
________________________________________________________________________________
Did you have any indication that nymphs contaminated themselves in addition to consumption of conditions?
Authors: Yes, mycosis is the indicator, growth out of the insect body.
______________________________________________________________________________
Reviewer 2 Report
Overall a standard study investigating the potential use of EPF's as control agents. The authors have carried out the experiments and analysed the data appropriately. Good source of literature cited within the introduction and discussion sections respectively. The study simply demonstrates the potential impact of EPF's against the grasshopper - I recommend publication of the ms. I have no issues with this study. It is well appropriate for 'Insects' and fit for publication. I only suggest that the authors do a thorough proof read of the ms - some typos and poor sentence structure occurs throughout.
Author Response
Responses to Reviewer 2
Overall a standard study investigating the potential use of EPF's as control agents. The authors have carried out the experiments and analysed the data appropriately. Good source of literature cited within the introduction and discussion sections respectively. The study simply demonstrates the potential impact of EPF's against the grasshopper - I recommend publication of the ms. I have no issues with this study. It is well appropriate for 'Insects' and fit for publication. I only suggest that the authors do a thorough proof read of the ms - some typos and poor sentence structure occurs throughout.
Authors: Changed were incorporated in the manuscript and typos and some sentences were corrected.